# Investigating the Effectiveness of a Carb-Free Oloproteic Diet in Fibromyalgia Treatment

**DOI:** 10.3390/nu16111620

**Published:** 2024-05-25

**Authors:** Giuseppe Castaldo, Carmen Marino, Mariangela Atteno, Maria D’Elia, Imma Pagano, Manuela Grimaldi, Aurelio Conte, Paola Molettieri, Angelo Santoro, Enza Napolitano, Ilaria Puca, Mariangela Raimondo, Chiara Parisella, Anna Maria D’Ursi, Luca Rastrelli

**Affiliations:** 1NutriKeto_LAB Unisa—“San Giuseppe Moscati” National Hospital (AORN), Contrada Amoretta, 83100 Avellino, Italymariangela.atteno@gmail.com (M.A.); ipagano@unisa.it (I.P.); aurconte@gmail.com (A.C.); paolamolettieri@gmail.com (P.M.); ilariapuca@libero.it (I.P.); mariangelaraimondo@gmail.com (M.R.); chiara.parisella@gmail.com (C.P.); 2PhD Program in Drug Discovery and Development, Department of Pharmacy, University of Salerno, Via Giovanni Paolo II, 132, 84084 Fisciano, Italy; cmarino@unisa.it (C.M.); enapolitano@unisa.it (E.N.); 3Department of Pharmacy, University of Salerno, Via Giovanni Paolo II, 132, 84084 Fisciano, Italy; magrimaldi@unisa.it (M.G.); asantoro@unisa.it (A.S.); 4National Biodiversity Future Center (NBFC), 90133 Palermo, Italy; mdelia@unisa.it; 5Department of Pharmacy, Scuola di Specializzazione in Farmacia Ospedaliera, University of Salerno, Via Giovanni Paolo II, 132, 84084 Fisciano, Italy

**Keywords:** fibromyalgia, ^1^H NMR metabolomics, very-low-calorie ketogenic diet, biomarkers

## Abstract

Fibromyalgia (FM), a chronic disease with a high incidence in women, poses a significant challenge for diagnosis and treatment, especially due to the absence of specific biomarkers and the multifaceted nature of its symptoms, which range from neuromuscular pain to mood disorders and intestinal dysbiosis. While diagnosis currently relies on rheumatological clinical evaluations and treatment options mainly focus on symptom management, FM seems to have possible links with systemic metabolic dysfunctions with a common inflammatory root. In this context, a new therapeutic avenue emerges: could a therapeutic nutritional approach be the missing piece of the puzzle? Indeed, diet therapies employed particularly for metabolic syndromes proved recently to be efficacious for correcting systemic dysmetabolism and a high number of chronic inflammation conditions. In particular, the very-low-calorie ketogenic diet (VLCKD) demonstrated therapeutic benefits in many disorders. In the present study, we aimed to investigate the specific effects of two dietary interventions, namely the oloproteic VLCKD and the low-glycemic insulinemic (LOGI) diet, on two groups of female FM patients (FM1 and FM2) over a 45-day period. Utilizing clinical and laboratory tests, as well as non-invasive NMR metabolomic analysis of serum, urine, and saliva samples, we sought to uncover how these dietary regimens impact the metabolic dysfunctions associated with FM.

## 1. Introduction

Fibromyalgia (FM) is a chronic disease estimated to affect 2–4% of the general population [1] and mainly affects women (61–90%) [2,3]. FM is characterized by a mixture of symptoms; among them, the most common are musculoskeletal pain and chronic fatigue, non-restorative sleep, morning stiffness, depression, anxiety [4], and gastrointestinal symptoms [5]. Although it has been extensively studied, the etiology of FM is almost unknown [6] and no biomarkers are currently available for diagnosis, which is based on the patient’s clinical history and assessment of Rome III criteria (American College of Rheumatology ACR criteria) [7,8,9,10]. Pharmacological therapy is symptomatic and uses several different drug categories, such as analgesics, muscle relaxants, and non-steroidal anti-inflammatories; moreover, the recommended treatment for FM is multidisciplinary and includes lifestyle measures and psychological support [1,2,4,5]. Sometimes, FM diagnosis is further complicated by comorbidities, i.e., diabetes, overweight, and disorders involving the neurological, gastrointestinal, endocrine, and immune apparatus.

The systemic nature of the disease, the absence of a gold-standard drug, and the palliative effect of many pharmacological treatments shed light on new avenues for exploration [10]. Could a therapeutic nutritional approach be the missing piece of the puzzle [11]? Indeed, diet therapies employed particularly for metabolic syndromes proved recently to be efficacious for correcting systemic dysmetabolism and many chronic inflammatory conditions [12,13].

The very-low-calorie ketogenic diet (VLCKD) is a nutritional regimen characterized by a drastic reduction in carbohydrate intake [14]. At a biochemical level, it causes an increase in blood ketones, shifting the energetic metabolism towards ketone metabolism [15]. Nutritional approaches based on VLCKD proved effective in treating obesity, metabolic syndrome (diabetes), neurological and autoimmune diseases, acne, polycystic ovary syndrome, and cancer [16,17,18,19,20,21,22,23,24]. We recently demonstrated that a four-week VLCKD nutritional treatment induced remission of symptoms and correction of the systemic metabolic condition in psoriatic patients [25]. VLCKD proved to be a transformative treatment, emphasizing the body’s natural ability to shift its energy metabolism and thus battle a range of complex diseases in addition to obesity [14].

In recent years, the fascinating realm of metabolomics has played a pivotal role in unraveling the mysteries behind various health conditions, shedding light on the intricate biochemical pathways at play [15]. By delving deep into the metabolic signatures present in biological fluids, researchers have gained insights into the underlying causes of diseases and discovered potential biomarkers that hold promise for enhancing diagnosis and treatment strategies. The relative ease of sample preparation, the capability to quantify metabolite levels, the large degree of experimental reproducibility, and the non-destructive nature of NMR spectroscopy have made it the favored platform for clinical large-scale metabolomic studies, enabling the simultaneous qualitative and quantitative identification of low-molecular-mass compounds in biological fluids [15].

This study aimed to evaluate how a nutritional regimen based on a ketogenic diet protocol named the oloproteic diet influenced the clinical parameters and the metabolic profiles of patients with a diagnosis of fibromyalgia. We enrolled 45 women and divided them into two groups: FM1, who followed a carb-free VLCKD (ooloproteic diet), and FM2, who followed a low-glycemic and insulinemic (LOGI) diet [26]. The oloproteic diet protocol includes a calculated protein content based on the patient’s ideal weight and a low or normal content of lipids, depending on the patient’s BMI [26]. The LOGI diet includes moderate (100–150 g daily) low-glycemic-index carbohydrate content, normal protein, and low/normal lipid content. After 45 days of dietary therapy, we evaluated the efficacy of these interventions using a comprehensive approach, including rheumatological and laboratory tests and NMR metabolomic analysis of blood serum, urine, and saliva. Our findings revealed a remarkable improvement in various clinical parameters across both dietary protocols. However, the oloproteic diet stood out, demonstrating a significant impact on FM symptoms that was possibly related to a rebalancing of the biochemical pathways associated with pain perception, as revealed by metabolomic data from all biological fluids. Through the lens of metabolomics, our study provides compelling evidence of the therapeutic potential of dietary interventions in managing FM symptoms. By elucidating the metabolic changes associated with these dietary protocols, we offer new insights into the mechanisms underlying FM and open the way for more personalized and effective treatment strategies.

## 2. Materials and Methods

### 2.1. Samples-Size Calculation

Sample-size calculation was performed using GPower 3.1 software [27].

Factors considered for the calculation of sample numbers were as follows: the test power parameter (1-β), which is the probability of rejecting the null hypothesis when there is a real difference and reflects the minimum difference between treatments (intervention group and control group) was set to 99%; the level of significance, α, or the probability of obtaining a statistically significant difference when there is no difference, was set to 5%; the effect size (f2) was set to 0.15. The power analysis estimated that 23 subjects are needed for each group to ensure a statistical power of 0.90 with an alpha = 0.05. Therefore, we initially recruited 50 patients, but 9 patients dropped out during the study.

### 2.2. Participants

We recruited 41 FM patients from the Rheumatology Clinic of the AORN “San Giuseppe Moscati” of Avellino. All were women diagnosed with primary FM. They were examined at t0 by two independent rheumatologists, and the diagnosis was made according to the ACR criteria [7]. FM patients were subjected to nutritional evaluation and clinical laboratory analyses at Nutriketo Lab (Laboratory of Nutritional Research, AORN Moscati of Avellino).

After the eligibility criteria had been confirmed and the informed consent signed, the participants were consecutively assigned to the intervention (FM1) or control group (FM2). Due to the nature of the study, the assignment to the FM1 or FM2 group was made according to randomized and controlled blind criteria for patients and rheumatologists, but not for nutritionists. In particular, all patients were given dietary recommendations generically defined as conforming to a low-calorie, low-carbohydrate diet. No explanations were given regarding the amounts of the single micro- and macronutrients in the diet or of the dietary supplements recommended to make the nutritional regimen safe. Each participant was assigned a code to guarantee the anonymity and confidentiality of the data (Appendix A). All demographic information is reported in Table 1.

### 2.3. Clinical and Laboratory Evaluation

Each patient in the FM1 and FM2 groups underwent anamnestic and rheumatological clinical evaluation at t0 and t45 [28]. The patients were followed regarding the standard FM symptoms to evaluate protocol compliance and symptoms/benefits.

Hematological and urinary laboratory analyses were carried out at t0 and t45. The results are reported in Appendix A. At the same times, rheumatological clinical evaluation and metabolomic analysis were carried out.

The rheumatological assessment was carried out according to the ACR 2010 criteria. Patients were invited to give a semi-quantitative estimation of FM comorbidities on a scale ranging between 0 and 3 (0: absence of symptom; 1, low symptom intensity; 2, medium symptom intensity; 3, high symptom intensity).

All patients were subjected to the same rheumatological assessment at both time 0 and t45 because the rheumatologists were unaware of the nutritional interventions performed by patients.

Then, the average of the impact of the symptoms for each group was calculated at two times (Appendix A). Clinical evaluation of anxiety and depression was performed using the Hamilton Anxiety Rating Scale and Hamilton Depression Rating Scale (HAM-A and HAM-D); a pathological state was associated with HAM-A > 17 and HAM-D > 21 [29].

### 2.4. Dietary Intervention and Assessment

The oloproteic diet is a very-low-carbohydrate ketogenic diet (VLCKD) lasting 45 days and inspired by the Blackburn diet (PSMF—protein-sparing modified fast). It includes minimal quantities of carbohydrates, normal protein content, and lipid content depending on BMI [28]. The prescribed caloric content ranges from about 20–35 Kcal/kg of ideal body weight calculated according to the Lanzola formula [30], with a higher caloric amount for patients with lower BMI. The protein quota administered was equal to 1.4 g of protein per kg of ideal body weight, with about 50% of the total protein quota derived from high-protein foods (meat, fish, eggs) and the other 50% derived from whey protein, amino acids, and hydrolyzed collagen supplements. The daily amount of carbohydrates was less than 10 g (considering the negligible carbohydrates contained in vegetables). In comparison, the lipid content ranged between 45 and 100 g of lipids (extra virgin olive oil and coconut oil). The diet included two meals daily, with all possible vegetables except those high in carbohydrates [26].

Herbal remedies with draining activity (orthosiphon and taraxacum) and alkaline salts (citrates and bicarbonates of potassium and magnesium) were used for replenishment of potassium and magnesium losses, which are expected in ketone-body metabolism.

The nutritional protocol administered to FM2 patients, the LOGI diet, was characterized by specific low calorie content based on the patient’s ideal weight and BMI. It included moderate low-glycemic carbohydrate content (100–150 g), 1.2 g biological protein intake per kg of ideal body weight [31], and low or normal lipid intake.

Patients were advised to drink at least 2 L/day of alkaline bicarbonate-calcium, low-sodium water. At enrollment, all treatments with oral hypoglycemic agents and diuretics were discontinued.

### 2.5. Sample Preparation for NMR Metabolomic Analysis

Blood sera, urine, and saliva were collected from all patients to perform NMR-based metabolomic analysis.

NMR sample preparation was performed as previously reported [30,32]. To obtain the blood serum sample, the whole blood was collected into tubes without anticoagulant and allowed to clot at room temperature for 30 to 120 min. After centrifugation at 12,000× *g*, the blood serum was aliquoted and stored at −80 °C in Greiner cryogenic vials before NMR spectroscopy measurements. Samples were thawed at room temperature before they were transferred to a 5 mm heavy-walled NMR tube. NMR samples were prepared by adding 200 μL of phosphate buffer to 300 μL of blood sera, including 0.075 M Na_2_HPO_4_ × 7 H_2_O, 4% NaN_3_, and H_2_O. trimethylsilyl propionic-2,2,3,3-d_4_ acid, sodium salt (TSP 0.1% in D_2_O) was used as an internal reference for the alignment and quantification of NMR signals; the mixture was homogenized by vortexing for 30 s and then transferred to a 5 mm NMR tube (Bruker® SampleJet NMR) before analysis started [32].

Saliva samples were collected in Sartstedt Salivette^®^ hygienic saliva-collection devices according to the standard operating procedure (SOP) for metabolomic-grade saliva samples [30]. After they were spun at a rotation speed of 3000 rpm, samples were stored at a temperature of −20 °C. Before they were transferred to a 5 mm heavy-walled NMR tube, samples were thawed at room temperature and 425 µL of each saliva sample was added to 25 µL of 1 M potassium phosphate buffer (pH 7.4) and 10 µL D_2_O. TSP 0.1% in D_2_O was used as an internal reference for aligning and quantifying the NMR signals [30].

Urine samples were prepared using 1.5 mL 24 h urine previously centrifuged at 15,000× *g* for 10 min at 4 °C to pellet any particulates in the sample; then, the urine was filtered using a 0.2 µm filter. Next, 500 μL of urine was transferred into a new tube and added to 50 μL of 50 mM phosphate buffer in 99.8% D_2_O [27,33].

### 2.6. NMR Data Acquisition

NMR experiments were conducted for all samples on a Bruker Ascend™ 600 MHz spectrometer equipped with a 5 mm triple resonance Z gradient TXI probe (Bruker Co., Rheinstetten, Germany) at 298 K. TopSpin version 3.2 was used for the spectrometer control and data processing (Bruker Biospin). 1D NOESY (Nuclear Overhauser Enhancement Spectroscopy) experiments were performed on saliva and acquired using 12 ppm spectral width, 19 k data points, with presaturation during relaxation delay and mixing time for water suppression and spoil gradient [34,35], 4 s relaxation delay, and mixing time of 10 ms. Carr–Purcell–Meiboom–Gill (CPMG) experiments were performed on serum and urine samples and acquired using 20 ppm spectral width and 32 k data points with f1 presaturation and T2 filter using D20 of 300 µs, D1 of 4 s. A weighted Fourier transform was applied to the time domain data with a line widening of 0.5 Hz followed by a manual step and baseline correction in preparation for targeted profiling analysis.

Due to acquisition problems, 7 samples were discarded from the second group, leaving 13 patients in that group.

### 2.7. NMR Data Processing

NMR spectra were manually phased and baseline-corrected. The R package ASICS was used to identify and quantify serum, saliva, and urine metabolites [36]. The ASICS package uses a method for automatically identifying and quantifying metabolites in the ^1^H NMR spectra. In the package, all phases of the analysis are combined (management of a reference library with spectra of pure metabolites, pre-processing, quantification, diagnostic tools to evaluate the quality of quantification, and post-quantification statistical analysis) [37].

### 2.8. Statistical Analysis

The *t*-test analysis was performed using MetaboAnalystR [38] to identify different clinical parameters [39]. Partial least-squares discriminant analysis (PLS-DA) was carried out with normalized metabolomics data using MetaboAnalyst 5.0 (http://www.metaboanalyst.ca, accessed on 10 April 2024) [40]. The performance of the PLS-DA model was evaluated using the coefficient Q^2^ (using the seven-fold internal cross-validation method) and the coefficient R^2^, which define the variance predicted and explained by the model, respectively. The loading plot was used to identify significant metabolites responsible for maximum separation in the PLS-DA score plot, and these metabolites were ranked according to their variable influence on projection (VIP) scores. VIP scores are weighted sums of squares of the PLS-DA weights that indicate the variable’s importance. The analysis of the pathways was carried out with the Metpa tool (included in R software (version 2.10.0)) [41]. Pathways with hit > 2, impact pathway > 0.45, and *p*-value and Holmp < 0.05 were considered significant [42].

Using the MetaboAnalyst 5.0 Enrichment analysis tool, we predicted localizations of organs, tissues, and subcellular compartments disrupted from the starting metabolomic dataset. Similarly, enzyme-specific dysmetabolism was predicted [43]. MVA considered FM1 at t0 vs. FM1 at t45 and FM2 at t0 vs. FM2 at t45. We considered significant organ-specific dysmetabolism with a hit value > 10 and *p*-value < 0.001, while a hit threshold > 4 and *p*-value < 0.01 were used for enzymatic prediction. A chord diagram was created using the package R cyclize 4.15 to better visualize the results [44]. For the construction of the diagram, only metabolites that could be used to discriminate between conditions according to the VIP value (performed comparing t0 vs. t45 for each group) were considered, and the impact of the metabolite on the tissue/organelle according to the *p*-value was evaluated.

## 3. Results

### 3.1. Clinical Analysis

Urine and blood biochemical analyses were performed (Appendix A). The absence of abnormal serum and urine parameters was used as a criterion for enrollment in the nutritional treatment.

Rheumatological clinical evaluations (WPI, SSS) were performed for FM1 and FM2 patients at t0 and t45. Table 2 reports the results of the ANOVA one-way test with post-hoc correction.

The average and standard deviation of the rheumatological score for each time point and each group according to the *t*-test are reported in Appendix A.

Figure 1A,B shows the violin histograms representing rheumatological score, WPI, and SSS. The histograms reported a similar value at t0 and reduced WPI and SSS scores in the FM1 group at t45. In detail, the SSS index at t45 decreased by 42.7% in FM1 and 25.33% in FM2. Concomitantly, WPI values decreased by 63,15% in FM1 and 25,67% in FM2.

In the analysis of general symptomatology, particular importance was given to the assessment of symptoms associated with anxiety and depression. According to the psychological tests, a pathological state was associated with HAM-A > 17 and HAM-D > 21 [29].

In FM1, the average value of anxiety symptoms decreased in the FM1 group at t45 (Figure 1C,D). In detail, we observed a reduction from 28.50 to 11.18 in FM1 and from 28.58 to 20.58 in FM2. Symptoms associated with depression changed from 17.89 to 14.89 in FM1 and from 19.21 to 12.27 in FM2.

Patients were also monitored from a nutritional point of view. Reductions of 6.32% (*p* value: 1.228 × 10^−8^) in weight and 2.43 (*p* value: 1.82 × 10^−11^) in BMI were observed in the FM1 group, while reductions of 5.45% (*p* value: 3.28 × 10^−5^) and 2.11 (*p*: 5.91 × 10^−5^) were observed in the FM2 Group.

### 3.2. Multivariate Data Analysis and Enrichment Analysis

NMR-based metabolomic analysis was performed by acquiring ^1^H NMR spectra for FM patients’ blood serum, urine, and saliva.

The original data matrices included metabolite concentrations calculated through quantitative resonance assignment of NMR spectra; therefore, multivariate analysis (MVA) was performed on the matrices, including 95 serum, 77 urine, and 59 salivary metabolite concentrations associated with FM1 and FM2 fibromyalgia patients at t45.

After normalization by median, log transformation, and media scaling, the data matrices were analyzed by supervised methods of partial least-squares determination analysis (PLS-DA) (Figure 2). The clustering validation was conducted by calculating the Q^2^ and R^2^ indices obtained with cross-validation methods (CV) (Appendix A). Figure 2 shows the score plots related to the serum, urinary, and salivary metabolomic profiles. A clustering of the profiles is observable for serum and urine in FM1 and FM2, suggesting that the VLCKD and LOGI nutritional regimens induce different changes in the metabolomic profiles of FM patients. On the other hand, the metabolomic profiles of FM1 and FM2 saliva show a partial overlap, indicating that in saliva biofluid, changes in the metabolome in response to VLCKD and LOGI nutritional treatments are not observable.

We performed VIP score analysis to identify the metabolites responsible for the changes in the metabolomic profiles, as shown in Figure 2A–C. Based on stringent criteria, only metabolites with a VIP value > 1.4 were considered significant.

The table in Figure 2A indicates that the FM1 treatment increased alanine and betaine concentrations at t45, while the FM2 treatment decreased concentrations of leucine, d-AMP, phenylalanine, arabitol, GABA, glutamine, tyrosine, butyrate, β-hydroxyisovalerate, allantoin, and glycogen.

VIP analysis of the FM1 urinary profile shows decreased concentrations of methylguanide, phosphocholine, beta-alanine, TMAO, creatinine, and citrulline (Figure 2B). In addition, high concentrations of saccaric, fucose, glycerol, sorbitol, myo-inositol, and glucose-6-phosphate are detected in the urine of the FM1 group compared to that of the FM2 group.

Furthermore, we observed higher concentrations of acetic acid, leucine, ornithine, and fucose in FM1 saliva than in FM2 saliva, whereas low concentrations of urea and glycolate were observed.

Moreover, VIP score analysis comparing t0 vs. t45 of each group was performed to assess the specific impact of different nutritional regimens on the metabolome (Table 3, Appendix A).

Considering the metabolites detected by VIP in Table 3, Enrichment Tissue/organelle analysis was performed to investigate the tissue/organelle localization of the unbalanced metabolites (Appendix A).

The chordiagram showed that alteration of the serum metabolomic profile of FM1 patients is consistent with dysmetabolism involving the mitochondria and the brain. In particular, such dysmetabolisms are consistent with reduced levels of N acetyl aspartate, ATP, and phenylalanine and an increase in levels of guanidoacetate. In addition, the dysregulation of the mitochondrial pathways is also consistent with an increase in levels of ornithine and malate (Figure 3A). On the other hand, the blood serum metabolomic profile of the FM2 group is consistent with liver and pancreas dysfunctions, as demonstrated by the increase in levels of leucine, acetoacetate, and glutamine concentrations and the reduction in levels of valine and alanine (Figure 3B).

The analysis of the metabolomic profile of urine revealed, in the FM1 group, the impact of the nutritional regime on the intestine, which was mainly observable in decreased concentrations of valerate, β-hydroxyalerate, isoleucine, and citrulline, which are considered biomarkers of intestinal dysbiosis (Figure 3C). By contrast, the urine metabolomic profile of FM2 patients confirmed the significant dysmetabolism of succinate, mannose, malate, glucose, and citrate, which is consistent with a significant effect on energetic pathways and heavy involvement of the pancreas (Figure 3D).

For the salivary profile of the FM2 group, it was not possible to construct the chordiagram, as the discriminating metabolites are represented only by glucuronic acid and ornithine; on the contrary, the FM1 group shows a predominant dysregulation of the liver, as demonstrated by the increased excretion of glycolate and sorbitol (Figure 3E).

We carried out pathway analysis to further investigate the significance of the data collected by NMR. The pathway analysis focused on the FM1 metabolomic profile revealed dysregulation in several serum and urine pathways related to neurotransmission, in particular, disruptions were observed in alanine, aspartate, *and glutamate metabolism; phenylalanine and tyrosine metabolism; and taurine and hypotaurine metabolism.* By contrast, in FM2, dysmetabolism is evident in energetic biochemical pathways related to the *synthesis and degradation of ketone bodies*. Saliva analysis showed no significant pathways in the FM2 at t45. Metabolomic analysis of FM1 saliva samples detected metabolic changes regarding the metabolisms of *(i) alanine, aspartate, and glutamate; (ii) taurine and hypotaurine* (Table 4). On the other hand, no significant changes were detected in the FM2 saliva profile.

Urinary, salivary, and serum metabolomic profiles were used to predict enzyme dysregulation (Appendix A). The Veen diagram shown in Figure 4 indicates three enzymes that, according to the metabolomic data of all the fluids, may have been modulated in the FM1 group after 45 days of nutritional treatment: *glutamine synthase, glyceraldehyde-3-phosphate, dehydrogenase, and trehalose exchange.* By contrast, for the FM2 group, the enzymatic prediction produced significant results only in serum, indicating dysregulation of the enzymes related to energy pathways such as *citrate synthase and pyruvate decarboxylase* (Appendix A).

## 4. Discussion

Fibromyalgia, a complex syndrome characterized by widespread muscle, bone, and nerve pain, disproportionately affects women [29]. Despite extensive research, its exact causes remain elusive, posing challenges to accurate diagnosis and effective treatment [40]. Current pharmacological approaches mainly offer symptomatic relief, prompting exploration of alternative therapies such as nutrition-based interventions.

In recent years, dietary strategies have garnered attention as promising avenues for managing fibromyalgia symptoms. Among these, low-calorie diets and low-fermentable-oligo-di-and-monosaccharides-and-polyols (FOD-MAP) diets have shown efficacy in alleviating pain and enhancing overall function [45,46,47,48,49]. Beyond pain relief, these dietary interventions have been linked to improvements in sleep quality, anxiety, and depression and reduction in inflammatory markers, thereby enhancing patients’ quality of life [16,17,18].

The very-low-calorie ketogenic diet (VLCKD) stands out as a nutritional approach marked by a significant reduction in carbohydrates and a proportional increase in protein and fat [14]. This dietary strategy triggers a metabolic shift, elevating blood ketone levels and directing energy metabolism towards utilizing ketone bodies [50]. Beyond its traditional use for weight management, VLCKD has demonstrated effectiveness in addressing a spectrum of inflammatory conditions, including metabolic syndrome, diabetes, neurological disorders, autoimmune diseases, acne, polycystic ovary syndrome, and even certain cancers [16,17].

In our recent investigation, employing NMR metabolomic analysis, we uncovered the therapeutic potential of VLCKD in managing psoriasis, showcasing symptom remission and a reduction in systemic inflammation [25]. Building upon these findings, our current study delves into the application of a specific VLCKD variant, known as the oloproteic diet, in the treatment of 22 female fibromyalgia patients over a 45-day period.

The oloproteic diet regimen is meticulously designed, featuring minimal carbohydrate intake, a protein content tailored to each patient’s ideal weight, and a moderate-to-low lipid intake based on BMI [26]. This dietary protocol is complemented with essential amino acids and high-quality proteins to ensure a balanced nitrogen intake [51]. In contrast, the FM2 patients received the LOGI diet, which is characterized by moderate consumption of low-glycemic-index carbohydrates, normal protein levels, and low/normal lipid content.

All patients undergoing the 45-day dietary intervention showed notable improvements in rheumatological and psychological symptoms, underscoring the significance of well-designed, low-carbohydrate nutritional approaches in managing FM symptoms. Remarkably, patients adhering to the oloproteic diet witnessed a remarkable reduction of over 50% in rheumatological WPI and SSS scores, alongside substantial improvements in psychological parameters such as HMA-A and HAM-D (Table 2, Figure 1). In contrast, those following the LOGI diet (FM2) experienced a less pronounced decrease, with no more than a 25% improvement in comparable parameters. This apparent discrepancy underscores the importance of tailoring dietary interventions to achieve optimal outcomes in FM symptom management.

To investigate the systemic effects of dietary treatments, we conducted an NMR-based metabolomic analysis of FM patients using blood sera, urine, and saliva. Our data show that both the oloproteic and LOGI dietary interventions brought about notable changes in the metabolomic profiles of FM patients, indicating beneficial effects on overall metabolism. However, while the impact of the LOGI diet seems to be predominantly on energy metabolism (Table 4) the metabolomic changes induced by the oloproteic treatment have a more systemic effect involving metabolic pathways possibly related to fibromyalgia-associated dysfunction.

Accordingly, patients after LOGI dietary treatment show a serum metabolomic profile characterized by increased levels of acetoacetate and the ketogenic amino acid leucine with decreased levels of the glucogenic amino acids valine and alanine [52] (Table 3). Moreover, their serum metabolome reveals significantly increased concentrations of guanidinoacetate and glutamine, both of which are activators of the tricarboxylic acid (TCA) cycle [53].

Patients after oloproteic dietary treatment also show increased concentrations of the metabolites responsible for the metabolomic change; nevertheless, their metabolome is marked by evident changes in metabolites not immediately related to ketogenic metabolism. Detailed examination of these metabolomic changes suggests that the significant regression of fibromyalgia symptoms—heightened pain sensitivity, anxiety, chronic fatigue, disrupted sleep, and morning stiffness—induced by the oloproteic diet is the result of several metabolic contributions: (i) rebalancing of amino acid metabolism, (ii) rebalancing of inflammatory conditions, (iii) increased dopaminergic transmission, (iv) modulation of GABAergic transmission, and (v) availability of additional energy sources to cope with neuromuscular stress conditions. 

i.Rebalance of amino acid metabolism involved in neurotransmission: The metabolome of FM1 patients at t45 shows decreased serum phenylalanine concentration and increased urinary isoleucine excretion. In keeping with these data, enrichment analysis revealed alteration of biochemical pathways responsible for synthesizing those amino acids that also have roles as neurotransmitters, specifically (i) alanine, aspartate, and glutamate, (ii) D-glutamine and D-glutamate, (iii) phenylalanine and tyrosine, and (iv) taurine and hypotaurine (Table 3). Exploration of the scientific literature in search of a correlation with fibromyalgia through which to interpret these data rapidly revealed studies supporting the association of neuropathic and muscular pain with the dysregulation of amino acid metabolism [2,54].ii.Rebalancing of inflammatory conditions. FM1 patients at t45 showed a significant decrease in levels of glucuronic acid (Table 3). This is known to be a ligand of toll-like 4 receptors that exacerbate inflammatory conditions and increase pain severity [33]. Accordingly, the oloproteic diet, in FM1 patients, seems to foster a rebalancing of inflammatory conditions that contribute to FM’s etiology. Indeed, previous scientific evidence showed elevated systemic levels of pro-inflammatory cytokines like IL-6 and IL-8 in FM patients compared to healthy individuals [34].iii.Based on specific tissue-organelle enrichment, the dysregulation of ketogenic amino acids and energy metabolites has important repercussions at the intestinal level. A significant proportion of our FM patients, exceeding 50%, experience intestinal dysbiosis [55,56], manifesting as symptoms such as dysentery or constipation, recurrent cystitis, and vaginal discharge (Appendix A). Confirming the effect of the oloproteic diet on alleviating inflammatory conditions in the gut, treatment with the oloproteic diet leads to notable improvements in these symptoms, with reductions of 59.1% in cystitis and 50.01% in vaginal discharge. Improvement in these symptoms was accompanied by a significant decrease in dysbiosis biomarkers in urine (including hydroxyvalerate, valerate, citrulline, and TMAO) [57,58,59] and saliva (glycolate and urea) [60,61] (Figure 2B,C, Appendix A).iv.Increase in dopaminergic transmission: Metabolomic data show that the metabolomic profile of FM1 patients at t45 is characterized by significantly decreased levels of tyrosine and phenylalanine. As tyrosine and phenylalanine are catecholamine precursors, their diminished levels are consistent with increased production of catecholamines [62] and, thus, increased catecholaminergic transmission. Previous scientific inquiries have revealed a link between abnormal pain perception in FM patients and the down-regulation of catecholamine transmission.v.Modulation of GABAergic transmission: VIP score analysis indicates that the serum metabolome of FM1 patients at t45 exhibits significantly decreased GABA and increased guanidinoacetate (GAA) concentrations. It is well known that fibromyalgia is associated with an alteration of GABAergic neurotransmission [53,63]. Indeed, several pharmacological therapeutic interventions make use of GABA inhibitors [64].vi.Availability of additional energy sources to cope with neuromuscular stress conditions: Our data show that blood sera of FM1 patients at t45 report an increase in *N*-acetyl aspartate. According to tissue-specific organelle enrichment, an increase in *N*-acetyl aspartate is consistent with neuromuscular-tissue dysmetabolism. Specifically, *N*-acetyl aspartate serves as a reservoir for glutamate [52,63] and acts as an energy source for cells during periods of stress when glucose, their primary fuel, is limited.

Metabolomic analysis of different biological fluids was performed to increase the analysis’s robustness and the data’s reliability. To attain the maximum results from our blood serum, urine, and saliva study, we performed enrichment analysis and enzymatic prediction [44] using MetaboAnalyst 5.0 (http://www.metaboanalyst.ca, accessed on 10 April 2024) [43]. The metabolites from all the biological fluids were compared with those deposited in the Human Metabolome Database (HMDB) to identify potential alterations in enzyme modulation or tissue dysfunctions that would be significantly consistent with the observed alteration in metabolite concentrations.

Accordingly, we found that restoring metabolic balance in the intestines of FM1 patients correlates with the modulation of the trehalose exchange enzyme (entry EC 2.3.1.122) (Figure 3, Appendix A). Intriguingly, genes encoding trehalose are associated with cell-wall homeostasis in response to stress and virulence of *Candida albicans* [65,66]. This finding sparks an interesting hypothesis regarding the potential critical role of C. albicans proliferation in the pathogenesis of fibromyalgia. Based on these insights, further investigations are warranted to confirm the involvement of C. albicans in FM and explore potential virulence markers for FM diagnosis.

## 5. Conclusions

We conducted a study involving 45 female fibromyalgia (FM) patients who underwent different nutritional interventions for 45 days. FM1 patients followed the oloproteic diet, which is characterized by minimal carbohydrate intake, with protein and lipid quantities tailored to individual BMI. In contrast, FM2 patients adhered to the LOGI diet, which contains moderate levels of low-glycemic-index carbohydrates.

Clinical assessments revealed improvements in FM symptoms with both diets, although more pronounced effects were observed with the oloproteic diet. Corresponding with symptom improvements, we noted distinct changes in metabolomic profiles from NMR analysis. Intriguingly, while the impact of the LOGI was predominantly on energy metabolism, the metabolomic changes induced by the oloproteic treatment had a more systemic effect, involving metabolic pathways possibly related to fibromyalgia dysfunction. Although these findings require validation in future studies, they preliminarily suggest that the oloproteic diet protocol could serve as an effective treatment for the significant reduction of FM symptoms. In contrast, the LOGI diet may be useful to sustain the benefits achieved with the oloproteic diet over time. 

## Figures and Tables

**Figure 1 nutrients-16-01620-f001:**
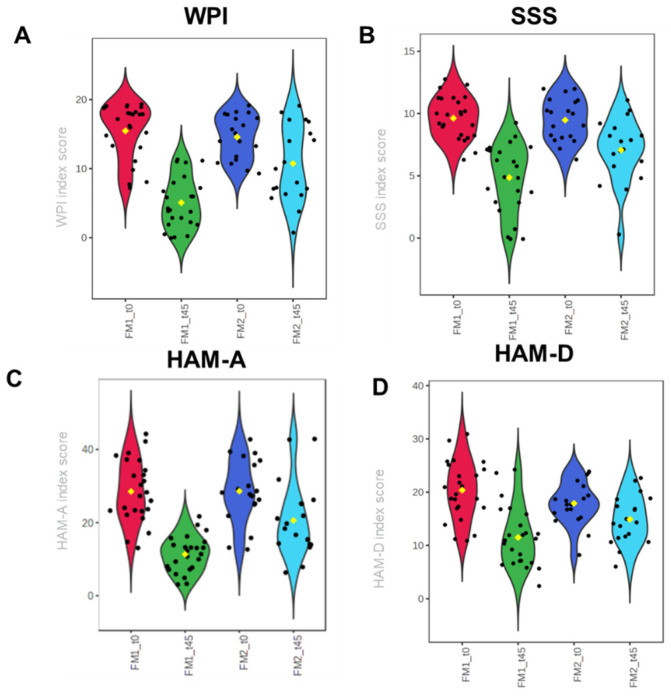
Violin graphs representing fibromyalgia patients’ rheumatological (**A**,**B**) and psychological scores (**C**,**D**) before and after nutritional treatment. The yellow diamond represents the average value.

**Figure 2 nutrients-16-01620-f002:**
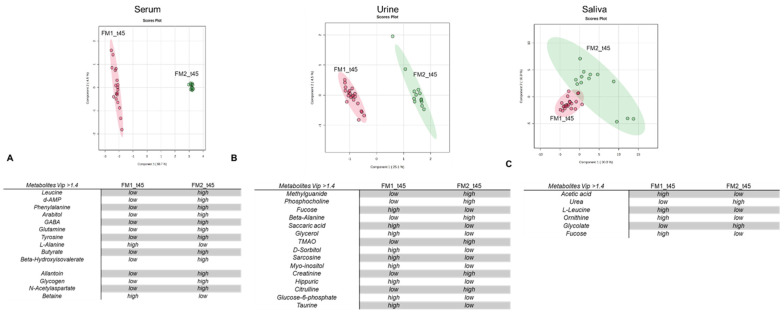
PLS-DA score scatterplot for the ^1^H NMR data collected in 1D-^1^H-CPMG spectra for serum and urine and 1D-NOESY for saliva acquired at 600 MHz. Data represent the serum, urine, and saliva profiles from FM1 and FM2 after 45 days of the diet. Metabolites discriminating between FM1 and FM2 according to VIP score analysis after 45 days (t45). Results are based on NMR metabolomic serum (**A**), urine (**B**), and saliva (**C**) analysis.

**Figure 3 nutrients-16-01620-f003:**
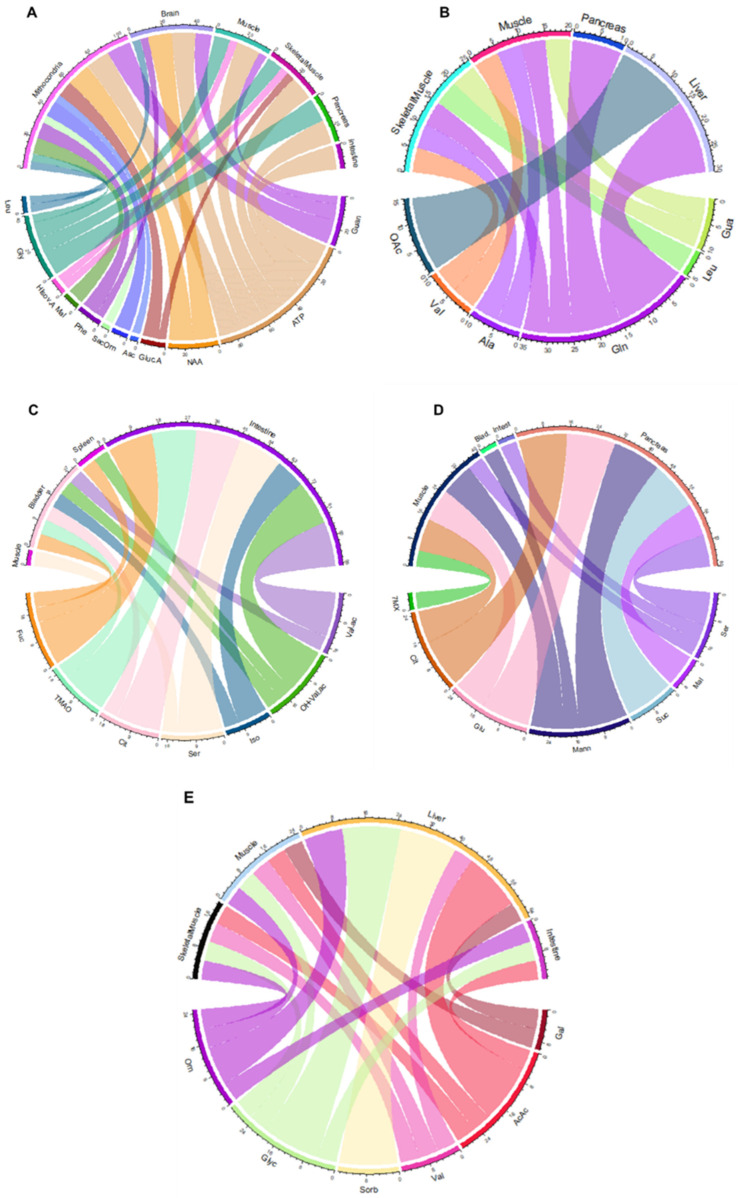
Chordiagrams related to enrichment tissue-organelle analysis were created for serum (**A**–**D**), urine (**B**–**E**), and saliva (**C**) profiles from FM1 and FM2 subjects, respectively, before and after the nutritional interventions. Leu: leucine; Gly: glycine; Hisov.A: hydroxyisovaleric acid; Mal: malate; Phe: phenylalanine; Sac: saccaric acid; Orn: ornithine; Asc: ascorbic acid; GlucA: glucoronic acid; NAA: *N*-acetyl-aspartate; Guan: guanidoacetate; Val: valine; Ala: alanine; Gln: glutamine; AcAc: acetoacetate. Fuc: fucose; TMAO: trimethylamine *N*-oxide; Cit: citrate; Ser: serine; Iso: isovalerate; OH-Val.ac: β-hydroxyisovalerate; Val.ac: valerate; Cit: citrate; Glu: glucose; Mann: mannose; Suc: succinate; Mal: malate; Ser: serine; glyc: glycolate; Sorb: Sorbitol; Gal: galactose.

**Figure 4 nutrients-16-01620-f004:**
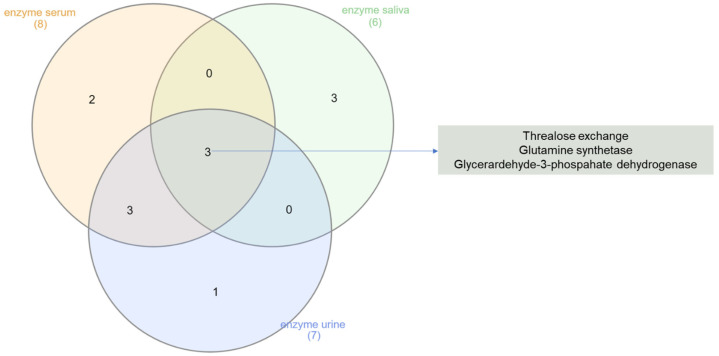
Veen diagram showing enzymes common predicted in common to be significantly dysregulated in serum, urine and saliva FM1 metabolomics profiles after 45 days of nutritional treatment.

**Table 1 nutrients-16-01620-t001:** Demographic patient information.

Parameters	Fibromyalgia First Group (N = 22)	Fibromyalgia Second Group (N = 19)
Sex (male/female)	0/22	0/19
Age (mean ± SD, year)	42.66 ± 8	40.22 ± 3
BMI (kg/m^2^)	31.82 ± 2.66	27.36 ± 7.12
Weight	78.00 ± 2.02	76.55 ± 1.52
Smokers (YES/NO)	5/17	3/16

**Table 2 nutrients-16-01620-t002:** ANOVA one-way results reported F value, *p*-value, False Discovery rate, and Fisher’s LSD to perform post-hoc test.

	f Value	*p* Value	FDR	Fisher’s LSD
WPI	30.555	2.33 × 10^−9^	9.31 × 10^−9^	FM1_t0–FM1_t45; FM1_t0–FM2_t45; FM2_t0–FM1_t45; FM2_t45–FM1_t45; FM2_t0–FM2_t45
SSS	23.568	4.27 × 10^−7^	7.44 × 10^−7^	FM1_t0–FM1_t45; FM1_t0–FM2_t45; FM2_t0–FM1_t45; FM2_t45–FM1_t45; FM2_t0–FM2_t46
HAM-A	23.231	5.58 × 10^−7^	7.44 × 10^−7^	FM1_t0–FM1_t45; FM1_t0–FM2_t45; FM2_t0–FM1_t45; FM2_t45–FM1_t45; FM2_t0–FM2_t47
HAM-D	13.414	3.33 × 10^−3^	3.33 × 10^−3^	FM1_t0–FM1_t45; FM1_t0–FM2_t45; FM2_t0–FM1_t45; FM2_t45–FM1_t45; FM2_t0–FM2_t48

**Table 3 nutrients-16-01620-t003:** Metabolites discriminating between FM1 and FM2 at t0 and after 45 (t45) according to VIP score analysis, Results are based on NMR metabolomic analysis of serum, urine, and saliva.

Serum	FM1		FM2	
Metabolites VIP > 1.7	t0	t45	t0	t45
guanidoacetate	low	high	/	/
ATP	high	low	/	/
N-acetylaspartate	high	low	/	/
d-glucoronic acid	high	low	/	/
ascorbic acid	low	high	/	/
ornithine	low	high	/	/
saccaric acid	low	high	/	/
phenylalanine	high	low	/	/
malic acid	low	high	/	/
β-hydroxyisovalerate	low	high	/	/
glycogen	high	low	/	/
l-leucine	low	high	low	high
acetoacetate	/	/	low	high
glutamine	/	/	low	high
alanine	/	/	high	low
valine	/	/	high	low
**Urine**	**FM1**		**FM2**	
**Metabolites VIP > 1.7**	**t0**	**t45**	**t0**	**t45**
valerate	high	low	/	/
β-hydroxyisovalerate	high	low	/	/
isoleucine	high	low	/	/
serine	high	low		
fucose	low	high	/	/
citrulline	high	low	/	/
TMAO	low	high	/	/
malate	/	/	high	low
succinate	/	/	low	high
mannose	/	/	low	high
glucose	/	/	low	high
7-methylxanthine	/	/	high	low
citrate	/	/	high	low
**Saliva**	**FM1**		**FM2**	
**Metabolites Vip > 1.7**	**t0**	**t45**	**t0**	**t45**
d-galactose	low	high	/	/
glycolate	high	low	/	/
acetoacetate	high	low	/	/
valine	high	low	/	/
sorbitol	low	high	/	/
d-glucoronic	low	high	high	low
ornithine	/	/	high	low

**Table 4 nutrients-16-01620-t004:** MetPa pathways results related to FM1 and FM2 biofluids. Common pathways are highlighted in the same color. Pathways with hits > 2; raw p and Holm adjust < 0.05 and impact > 0.45 were considered significant.

Serum FM1 t0_t45	Hits	Raw p	Holm Adjust	Impact
alanine. aspartate and glutamate metabolism	11	6.68 × 10^−6^	3.21 × 10^−5^	0.78
Arginine biosynthesis	8	8.36 × 10^−5^	3.57 × 10^−3^	0.6
D-glutamine and D-glutamate metabolism	3	8.53 × 10^−4^	2.39 × 10^−2^	0.5
Phenyalanine and tyrosine metabolism	3	1.39 × 10^−3^	3.20 × 10^−2^	1
taurine and hypotaurine metabolism	3	1.31 × 10^−2^	2.24 × 10^−2^	0.65
TCA	4	1.22 × 10^−2^	2.20 × 10^−2^	0.47
Synthesis and degradation of ketone bodies	4	1.99 × 10^−2^	1.08 × 10^−2^	0.60
**Urine FM1 t0_t45**	Hits	Raw p	Holm adjust	Impact
Aminoacyl-tRNA biosynthesis	14	3.90 × 10^−18^	1.8790 × 10^−17^	0.52
valine. leucine and isoleucine biosynthesis	5	1.4290 × 10^−14^	6.5290 × 10^−13^	0.75
Arginine biosynthesis	6	1.0390 × 10^−10^	4.1190 × 10^−9^	0.56
alanine. aspartate and glutamate metabolism	6	1.73 × 10^−4^	4.1590 × 10^−3^	0.53
taurine and hypotaurine metabolism	3	1.05 × 10^−3^	0.0285	0.71
**Saliva FM1 t0_t45**	Hits	Raw p	Holm adjust	Impact
alanine. aspartate and glutamate metabolism	6	3.67 × 10^−3^	1.3690 × 10^−1^	0.45
glycine. serine and threonine metabolism	9	5.73 × 10^−4^	0.0143	0.68
**Serum FM2 t0_t45**	Hits	Raw p	Holm adjust	Impact
Synthesis and degradation of ketone bodies	4	9.3490 × 10^−11^	0.00224	0.6
TCA	4	3.2290 × 10^−10^	2.0190 × 10^−11^	0.49
beta-alanine metabolism	3	0.005	0.00454	0.45
**Urine FM2 t0_t45**	Hits	Raw p	Holm adjust	Impact
Synthesis and degradation of ketone bodies	4	3.04 × 10^−4^	1.25 × 10^−2^	0.6

## Data Availability

The original contributions presented in the study are included in the article/Appendix A, further inquiries can be directed to the corresponding authors.

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
