# Peer review of "Investigating the Effectiveness of a Carb-Free Oloproteic Diet in Fibromyalgia Treatment"

_nutrients, 2024, doi:10.3390/nu16111620_

Round 1

Reviewer 1 Report (New Reviewer)

Comments and Suggestions for Authors

Thank you for the opportunity to review this manuscript. 

This is a blinded controlled clinical trial evaluating the effects of two dietary interventions on clinical and metabolomic measures of a cohort of women with fibromyalgia. The Authors compare Very-Low-Calorie Ketogenic Diet (VLCKD) to Low Glycemic Insulinemic (LOGI) diets adopted for 45 days, showing improvement in all clinical measures, as well as diet-specific changes in serum- urine- and saliva metabolomics. DA molecules are compared between baseline and 45 days in the two groups. Then, pathway analysis is performed and enzymatic dysregulations are inferred. 

The study is well-designed, and the manuscript is well-written. The question at hand is important and adds to a growing body of evidence highlighting diet as an important therapeutic measure in FM.

However, there are several concerns regarding the presentation of the results and their interpretation.

Major comments:

1.     Results

a.     Was there any follow-up regarding participant adherence to the diets? Was the overall caloric intake and the intake of macro- and micro-nutrients analyzed? 

b.     Please provide pain intensity before and after treatment.

c.      Table 2 – please present the average and standard deviation for each time point and each group. Also, please provide individual p values for each comparison (FM1 t0 vs. t45, FM2 t0 vs. t45, FM1 vs. FM2 t0, FM1 vs. FM2 t45).

d.     There seems to be a high variability in BMI – was the clinical response associated with baseline BMI?

e.     Was weight monitored during the study? What were the changes?

f.      Figure 2 – ordination analysis of serum, urine and saliva metabolomics. Please also provide t0 to demonstrate the baseline features of the two groups and the temporal changes.

g.     Figure 3 and Line 335 – most observed changes are in amino acids and short-chain fatty acids. These are often observed in metabolomic analyses and their attribution to liber or pancreatic dysfunction is over-simplified. Multiple factors determine their serum concentration, including uptake in the gut, excretion, tissue turnover, enzymatic cleavage and bacterial metabolism. 
The output of MetaboAnalyst cannot be presented in its raw format and must be analyzed to make biological sense of it. 

2.     Discussion

a.     There is some confusion between results (e.g. clinical measures, metabolite concentrations) and their interpretations (e.g. pathways to which metabolites belong, their putative functions and inferred changes in upstream enzymes).
I would encourage the authors to separate results from inferences to make it easier for the reader to notice the difference between what is observed and what is inferred. 
E.g: Line 446 – changes in serum metabolomics were primarily in amino acids, which makes sense given the change in protein intake. While AAs are important in the synthesis of neurotransmitters, they are also involved in numerous other host metabolic pathways. Thus, in the absence of measurable changes in neurotransmitters, this discussion is highly speculative.

Minor comments:

1.     Introduction

a.     Line 41 – the prevalence of FM is arguably higher. Please reference more up-to-date publications.

b.     Line 42 – while FM is more prevalent in women the number 90% is obsolete and was an artifact of older diagnostic criteria. The current estimation is around 3:1 women:men. 

c.      Line 51 – tender point stimulation is no longer a part of the clinical diagnosis of FM.

d.     Line 54 – the recommended treatment for FM is multidisciplinary, and includes lifestyle measures, medications and psychological support.

e.     Line 57 – The reference does not support the statement. There is ample evidence for inflammation in FM (e.g. autoantibodies, activated neutrophils). The paper on comorbid fibromyalgia describes the increased prevalence of FM in rheumatic and other diseases, but does not have mechanistic inference.

f.      Lines 81-88 – please mention studies directly looking at metabolomics in FM (e.g. Clos-Garcia, 2019, Minerbi 2019).

g.     Lines 100-109 – consider moving to the discussion.

2.     Methods

a.     Line 115 – when calculating the power analysis, what was the major outcome variable? What is its expected mean and STD? What is the expected between-group difference?

b.     Why was LOGI chosen as control?

c.      Line 120 – the reference is for RA diagnostic criteria, not FM.

3.     Results

a.     Is there any data on the baseline diet consumed by patients before the intervention?

b.     Supplementary Table 2 – did 100% of patients have skin candidiasis? 100% awakening refreshed? Please double-check these and other figures in the table.

4.     Discussion

a.     Both diets showed significant clinical value. In addition to providing between-diet statistical comparison, what makes VLCKD diet different from the LOGI diet and other low-carb diets? 

b.     Line 422 – Did all participants show clinical improvement? This is extraordinary for clinical trials and should be discussed.

c.      Line 456 – Ref 12 is irrelevant to the statement and ref 59 is outdated. Please revise.

d.     Line 464 – Serum GABA is not equivalent to synaptic GABA.

e.     Line 472 – please reference.

f.      Line 498 – why is a decrease in the IL4 ligand necessarily indicative of inflammation? It could also be the other way around (e.g. downregulation).

g.     Line 505 – the references are irrelevant to dysbiosis in FM. Did you mean IBS? Other studies show gut microbiome changes in FM (e.g. Clos-Garcia 2019, Minerbi 2019).

h.     Line 506 – why are cystitis and vaginal discharge attributable to gut dysbiosis? Please reference.

i.       Lines 507-510 – There are far more accurate ways to demonstrate dysbiosis. In fact, changes in AAs and SCFAs are highly associated with gut microbiome changes.

j.       Line 511-519 – this analysis is highly speculative and unsupported by the data ot references. Consider revising.

Author Response

we attached a rebuttal

Reviewer 2 Report (New Reviewer)

Comments and Suggestions for Authors

This is an interesting study. I have several comments. 

First, the introduction is too fragmented. I suggest to concentrate them into three paragraphs. 

Second, what is the hypothesis of the present study? The authors need to state. 

Third, in the method portion, the section of sample size estimation should be seprated from other parts. 

Fourth, to sepate the participants into FM1 and FM2 groups is quite confusion. Why not use "FM" and "CON" groups? 

Fifth, there is a lack of flow diagram of the whole research process. The authors should provide. 

Author Response

we attached a rebuttal

Round 2

Reviewer 2 Report (New Reviewer)

Comments and Suggestions for Authors

The article has been well revised. 

This manuscript is a resubmission of an earlier submission. The following is a list of the peer review reports and author responses from that submission.

Round 1

Reviewer 1 Report

Comments and Suggestions for Authors

This randomised control trial of a reduced carbohydrate diet versus a low glycaemic index diet in individuals with fibromyalgia is an area of clinical interest and need. However, the study has various flaws and, as it is written, I am not convinced of the ethical nature of the dietary interventions delivered. Detailed comments below:

Introduction

The Introduction is comprehensive in its explanation of fibromyalgia and aetiology / treatment options. It is, however, rather long. I would suggest reducing the length, for example, by moving some of the paragraph commencing ‘In a pilot study’ (and also some parts of the following 3 paragraphs) to the Discussion. The last 3 paragraphs of the Introduction could also go in Methods and/or Results.

Please provide more of a background to the ‘Oloproteic Diet®’: it has a registered trademark, so is this an industry-adopted diet?

Page 3 lines 103-4: ‘euphoric and tonic effects’. What does this mean and what peer-reviewed studies are these terms based on?

Methods

How were patients blinded to their intervention group? It seems as if participants were all oral eaters, so how was this possible? How was randomisation allocated? Sample size calculation?

‘Standard’ laboratory analyses – please clarify. Do you mean according to routine clinical practice in that particular centre? And also please clarify what is meant by ‘Anamnestic and rheumatological clinical evaluation’.

‘herbal remedies’ (page 4 line 167) – what do you mean by this? Were participants not supplemented with a multivitamin/mineral supplement?

Please provide more information about the LOGI. Participants had 100-150g carbohydrates, but were these all low-glycaemic index foods? What advice were participants given in order to follow this diet?

Results

Did rheumatological and psychological test results differ between the two randomisation groups?

Page 9 lines 315+ This paragraph would be better placed in the Discussion.

The study is labelled as using a ‘very low carbohydrate ketogenic diet’. Did the participants randomised to the Oloproteic diet produce ketones? I would be surprised if so, considering the amount of protein prescribed.

Discussion

Please start the discussion with a brief summary of your findings. And then go on to compare your findings to similar studies in the literature. As it stands, in the first few paragraphs of the Discussion, it is not clear what belongs to this study and what belongs to other articles.

Line 445: ‘While Oloproteic® diet significantly impacts neurotransmission’ – how is this statement supported by the results?

Comments on the Quality of English Language

Generally sound, but some editing required.

Author Response

we attached the file

Reviewer 2 Report

Comments and Suggestions for Authors

TO AUTHORS:

The paper of Castaldo et al. describes the effects of VLCKD and and LOGI diets on two groups of female patients. This observatory study fits well into the scope of this journal and is well written. However, I recommend a major revision of the manuscript, as there are several significant concerns that need to be addressed.

ABSTRACT:

Abstract is well written.

INTRODUCTION:

Introduction is well written and easy to follow.

MM

The experimental protocols are well designed and written clearly and in detail, except statistical analysis, which should be rewritten according to comments in the results section (see below).

RESULTS

MAJOR COMMENTS:

·         Graphical displays should be clearly visible. Please increase font and size of the graphs.

·         Figure 1 When statistically analyzing the data described in Figure 1, the authors should consider a test other than Student’s t-test, i.e. a two-way ANOVA with the appropriate post hoc test. The reason for using such a test is to examine the simple or interaction effect of two categorical variables on the dependent variable, as is the case in Figure 1. A two-way ANOVA is therefore the method of choice for drawing the most accurate conclusion possible regarding the effect of each variable (time or intervention) and should be performed. Any statistical significance between groups should be placed into the graphs with the appropriate text below each graph (in legends to figures) explaining the statistical method along with significances and number of samples used for each graph. Also, you should put the title into y axis as the reader must know what is measured in the graph.

·         Lines 466-467 - which part of the paper does this sentence refer to?

DISCUSSION

·         Page 12, line 382: please do not begin the sentence with a numerical value. In addition, these sentences (line 382-387) are not necessary in the discussion section.

Lines 388-398. Here the authors again repeat their own results. Instead, the authors should focus on interpreting the results in the context of existing literature, emphasizing the significance of their findings.

CONCLUSION

The authors should be cautious with statements such as improvement of mitochondrial function, antioxidant status, etc., as they do not have other analyses besides metabolomic analysis, such as real-time PCR, Western blot analysis and analyses that would determine the specific activities of individual enzymes involved in certain parameters. this limitation of the study must be mentioned in the conclusion.

MINOR

Page 1, line 42: please change “gender disease” with “gender-related” disease.

Please pay attention to the extra space between words and delete it accordingly.

Author Response

we attached the file

Round 2

Reviewer 2 Report

Comments and Suggestions for Authors

The authors have answered most of the points adequately. However, there is still one important point in the study concerning the appropriate statistical analysis, which the authors have not provided. Instead of a two-way ANOVA, the authors performed a one-way ANOVA with a post-hoc test, which should be more conservative. Although the one-way ANOVA is not wrong per se, I think that for this type of data a two-way ANOVA would be a far more appropriate analysis. I therefore strongly recommend that the authors present a two-way ANOVA, as this is the most appropriate method for this type of data. Please also explain the statistics in the graphical representations in the legends to the figures. These points need to be considered in a revised manuscript.

Author Response

We thank the reviewer for his her comment. We modified the manuscript in the clinical results section as suggested.
